# Estimating the undetected emergence of COVID-19 in the US

**Emily M. Javan** [1]* , **Spencer J. Fox** [1,2] , **Lauren Ancel Meyers** [1,3]

**1** Department of Integrative Biology, University of Texas at Austin, Austin, TX, United States of America,
**2** Department of Epidemiology & Biostatistics and Institute of Bioinformatics, University of Georgia, Athens, GA, United States of America, **3** Santa Fe Institute, Santa Fe, New Mexico, United States of America

☯ These authors contributed equally to this work.
* emjavan@utexas.edu

**Data Availability Statement:** All county case data are available from the New York Times GitHub repository, https://github.com/nytimes/covid-19-data, and model parameters values are derived or cited within the paper.

## Abstract

As SARS-CoV-2 emerged as a global threat in early 2020, China enacted rapid and strict lockdown orders to prevent introductions and suppress transmission. In contrast, the United States federal government did not enact national orders. State and local authorities were left to make rapid decisions based on limited case data and scientific information to protect their communities. To support local decision making in early 2020, we developed a model for estimating the probability of an *undetected* COVID-19 epidemic (epidemic risk) in each US county based on the epidemiological characteristics of the virus and the number of confirmed and suspected cases. As a retrospective analysis we included county-specific reproduction numbers and found that counties with only a single reported case by March 16, 2020 had a mean epidemic risk of 71% (95% CI: 52–83%), implying COVID-19 was already spreading widely by the first detected case. By that date, 15% of US counties covering 63% of the population had reported at least one case and had epidemic risk greater than 50%. We find that a 10% increase in model estimated epidemic risk for March 16 yields a 0.53 (95% CI: 0.49–0.58) increase in the log odds that the county reported at least two additional cases in the following week. The original epidemic risk estimates made on March 16, 2020 that assumed all counties had an effective reproduction number of 3.0 are highly correlated with our retrospective estimates (r = 0.99; p<0.001) but are less predictive of subsequent case increases (AIC difference of 93.3 and 100% weight in favor of the retrospective risk estimates). Given the low rates of testing and reporting early in the pandemic, taking action upon the detection of just one or a few cases may be prudent.

## Introduction

The COVID-19 (coronavirus disease of 2019) pandemic claimed over 350,000 American lives in 2020 [1]. Early in the pandemic, when confirmed case counts were still relatively low across the US, the federal government left decision making largely to state and local public authorities. Amidst great uncertainty, leaders faced the unprecedented challenge of balancing the threat of a mostly undetected but deadly virus against the economic and societal costs of shelter-in-place and travel restrictions. At the time, most COVID-19 cases were not reported given

**Funding:** EMJ, SJF, and LAM acknowledge the financial support from National Institutes of Health (NIH), grant no. U01 GM087719 and project no. 5T32LMO012414. The NIH played no role in study design, data collection and analysis, decision to publish, or preparation of the manuscript. https://www.nih.gov/.

**Competing interests:** The authors have declared that no competing interests exist.

the high proportion of mild and asymptomatic infections, limited laboratory testing capacity and strict requirements for receiving tests (e.g. travel or contact with someone from Wuhan, China) [2, 3]. The CDC estimated that only one in ten COVID-19 infections were reported during the early phase of the pandemic [4].

As the first cases of COVID-19 were reported, decision makers urgently needed to determine whether they reflected sporadic clusters stemming from recent introductions or sustained community transmission that might evolve into a large epidemic. In the southern US, the 2016 expansion of Zika Virus (ZIKV) across the Americas posed a similar challenge. Cryptic transmission meant that by the time a few cases were reported, a large epidemic could already be underway [5]. Here, we describe a stochastic susceptible-exposed-infected-recovered (SEIR) compartmental model framework for estimating the probability of an on-going, undetected epidemic (*epidemic risk*) from scarce case data. In this study, we use the term *epidemic* to refer to the county-level reproduction number of SARS-CoV-2 being greater than one. This is the threshold between self-sustained epidemic growth versus stuttering chains of transmission [6].

The approach was originally developed to support situational awareness for ZIKV and adapted for COVID-19. We apply it to estimating the risk of undetected COVID-19 epidemics in US counties during the emergence phase of the pandemic in early 2020. We present results from a model using the best estimates for COVID-19 epidemiological characteristics as of December 2022 (retrospective) and compare those results with those made in early March 2020 (original).

## Results

We modeled the stochastic emergence of COVID-19 accounting for county-specific transmission risks, potential superspreading events, asymptomatic infections, and disease-specific epidemiological characteristics (Table 1). We assumed county-specific transmission rates ranging from $R_e$ of 1.4 to 4.4 with a median of 2.8 as estimated in [7]. Based on the underlying $R_e$ and a 10% case detection rate [4], the chance that a county had an underlying COVID-19 epidemic (epidemic risk) was 7–28% with no detected cases, 51–85% upon the detection of a single case, and over 99% by the time 11 or more cases were detected under scenarios without non-pharmaceutical interventions in place (Fig 1). For example, Travis County, TX (the primary county representing the city of Austin, TX) was estimated to have an $R_e$ of 2.0 [7]. Our model estimates a 95% epidemic risk for Travis County based on the four cumulative cases reported by March 13, 2020, which increases to 99% on March 20, 2020 when there were twenty-one cumulative reported cases (Fig 1).

By March 16, 2020, counties reported between 0 and 489 cumulative cases totaling 4,009 nationally [16]. We estimate a national mean epidemic risk of 25% (95% CI: 11–99%) on that day and that epidemic risk exceeded 50% for roughly 15% of the 3,142 counties representing 63% of the US population (Fig 2A). By April 13, 2020, total reported cases in the US climbed to 467,158 and we estimate a mean epidemic risk of 82% (95% CI: 12–100%) with an estimated 85% of counties representing 96% of the US population having over a 50% epidemic risk (Fig 2B). Projected risks are generally higher for both larger transmission rates (Fig 1) and lower assumed case detection rates (S1 Fig). For the median $R_e$ = 2.8, the expected time between the first COVID-19 case report and the epidemic reaching 1,000 cumulative infections was 3.4 (95% CI 2.0–7.3) weeks. Waiting to act until the tenth reported case would shrink the time until 1,000 cumulative infections by 55% to 1.5 (95% CI 1.0–2.9) weeks (Fig 3).

To validate our model, we compare the estimated epidemic risk on March 16, 2020 to increases in reported case counts in the following week (Fig 4A). We find that our estimates of

**Table 1. Model parameters used for simulating COVID-19 outbreaks.**

| Parameter | Description | Original | Source | Retrospective | Source |
|---|---|---|---|---|---|
| $R_e$ | Effective reproduction number: Average number of new cases from one infected individual in a susceptible and non-susceptible population | 1.5 | [8] | Contiguous US: fit to each county | [7] |
| | | 1.1, 3.0 | [9] | Alaska and Hawaii: mean of urban-rural designation | [10] |
| $T_G$ | Generation time (days): Average length of time between consecutive exposures $$T_G = \frac{e}{v} + \left(\frac{1}{2}\right)\frac{n}{\delta} = T_E + \left(\frac{1}{2}\right)T_I$$ | 6 | [11] | 6 | [12] |
| $T_E$ | Latent period (days) | 1.25 | Fit to $T_G$ | 2.9 | [13] |
| $T_I$ | Infectious period (days) | 9.5 | [11] | 6.2 | Fit to $T_G$ |
| $e$ | Number of exposed compartments in boxcar implementation (min days of exposure) | 1 | floor($T_E$) | 2 | floor($T_E$) |
| $n$ | Number of infectious compartments in boxcar implementation (min days of infectiousness) | 7 | [11] | 6 | floor($T_I$) |
| $v$ | Latency rate: Daily probability of progressing from one exposed compartment to the next | 0.80 | $e/T_E$ | 0.69 | $e/T_E$ |
| $\delta$ | Recovery rate: Daily probability of progressing from one infectious compartment to the next | 0.73 | $n/T_I$ | 0.97 | $n/T_I$ |
| $\eta$ | Daily detecting rate: The daily probability of an infectious individual being detected, $\frac{0.1}{T_I}$ | 0.01 | [14] | 0.01 | [14] |
| $k$ | Total dispersion parameter of negative binomial distribution | 0.16 | [15] | 0.16 | [15] |
| | Daily detection rate: Probability of an on-going infection becoming a reported case | 0.1 | [4] | 0.1 | [4] |
| | R code for number of new infectious individuals drawn daily: $$rnbinom\left(n = 1, prob = \frac{k}{R_e + k}, size = \frac{k}{T_I}\right)$$ | | | | |

epidemic risk correlate significantly with the probability that a county reported additional cases in the subsequent week (logistic regression, p<0.001). A 10% increase in estimated risk corresponds to an increase in the log odds of a county detecting at least one, two, or five new cases of 0.55 (95% CI 0.49–0.61), 0.53 (95% CI 0.49–0.58), and 0.57 (95% CI 0.53–0.61), respectively (Fig 4B and S2 Fig).

As additional validation of the modeling framework, we compare the estimates originally made in March 2020, before we had county-specific estimates of reproduction numbers (Table 1, Fig 5). At that time, we assumed all counties had the same effective reproduction numbers, ranging from 1.1 to 3.0; we also originally assumed a latent period of 1.25 rather than 2.9 days. Our original estimates assuming $R_e$ = 3.0 most closely match the retrospective estimates (Pearson's product-moment correlation, r = 0.99; p<0.001). Our original county-level risk maps (S3–S5 Figs) and estimates for the time until counties will reach 1,000 cumulative infections (S6 and S7 Figs) are also consistent with our retrospective analysis. Finally, our original estimates reliably predicted subsequent county case increases (S9 Fig). For example, assuming $R_e$ = 3.0, a 10% increase in estimated epidemic risk corresponds to an increase in the log odds of a county detecting at least one, two, or five new cases by March 23 of 0.48 (95% CI: 0.43–0.53), 0.49 (95% CI: 0.45–0.53), and 0.55 (95% CI: 0.51–0.59), respectively. Comparing logistic regression models built on the retrospective analysis to the original epidemic risk estimates where $R_e$ = 3.0, we find that the retrospective risk estimates more accurately predict the probability of a county reporting at least two new reported cases in the week following March 16, 2020 (AIC difference of 93.3 and 100% weight in favor of the retrospective risk estimates).

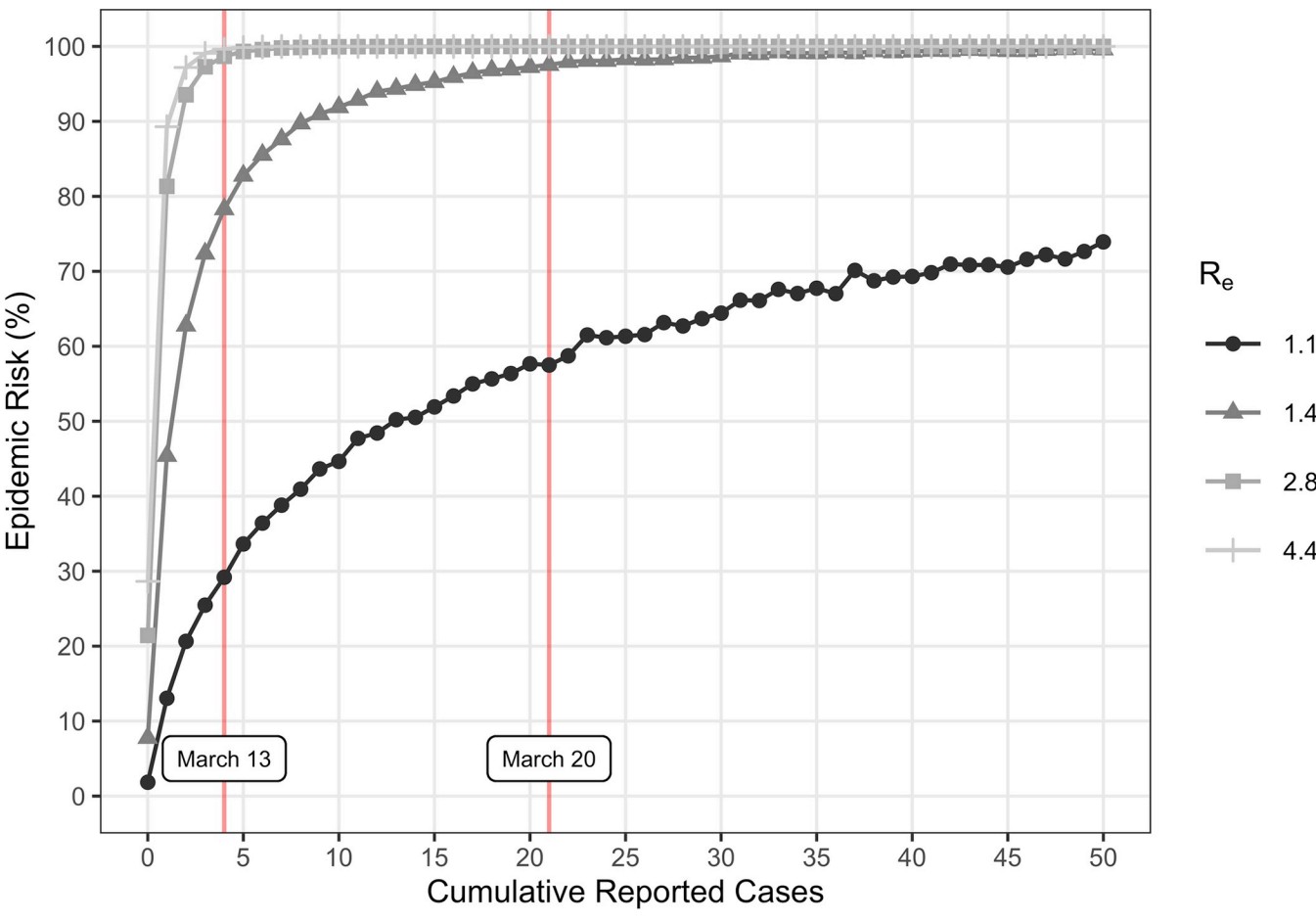

**Fig 1. Epidemic risk for the effective reproduction numbers ($R_e$) corresponding to reduced risk (1.1) and the minimum (1.4), median (2.8), and maximum (4.4) estimated across all US counties.** For a given number of reported cases, epidemic risk increased with estimated $R_e$. Epidemic risk is the percent of 100,000 simulations, for each $R_e$, that become epidemics. We classified a simulation as an epidemic if it reached 2,000 cumulative infections and had a minimum prevalence of 50 new infections per day. By the time a single case was reported, there was a 13%, 45%, 81%, or 89% chance of an ongoing epidemic for an $R_e$ of 1.1 (reduced risk), 1.4 (minimum), 2.8 (median), or 4.4 (maximum), respectively. County-specific risk was estimated from these curves. For example, Travis County, TX (red lines) had an $R_e$ of 2.0, which corresponds to an epidemic risk of 95% on March 13, 2020 and 99% on March 20, 2020 based on cumulative reported case counts of four and twenty-one on those dates, respectively. If the $R_e$ was instead estimated as 1.1 in Travis County, then the estimated risk would decrease to 57% based on the twenty-one cases reported on March 20. The model assumed a 10% case detection rate, generation time of 6.0 days, a latent period of 2.9 days, and infectious period of 6.2 days (Table 1 - retrospective).

## Discussion

The timing and rate of COVID-19 emergence varied widely across the US [17]. The earliest of the 3,142 US counties to report a case was Snohomish, Washington on January 21, 2020. By the first of March, April and May, 1%, 70% and 90% of all counties had reported at least one case, respectively. Using county-specific transmission risk estimates ($R_e$) and a 10% case detection rate, we estimate that by the time a county reported its first case it had at least a 50% chance of a growing epidemic. On March 16, 2020 mean risk was 25% (95% CI: 11–99%), and by April 13, 2020 risk exceeded 90% in 67% of counties containing 94% of the US population.

Our framework was developed in the first months of the COVID-19 pandemic to provide local situational awareness for both government officials and the public, at a time when the data were highly uncertain. On April 3, 2020, the New York Times published our first set of estimates as a national risk map appearing on the front page [18], which reached an estimated 694 million unique viewers according to Meltwater [19]. The map estimated that 70% of 3,142

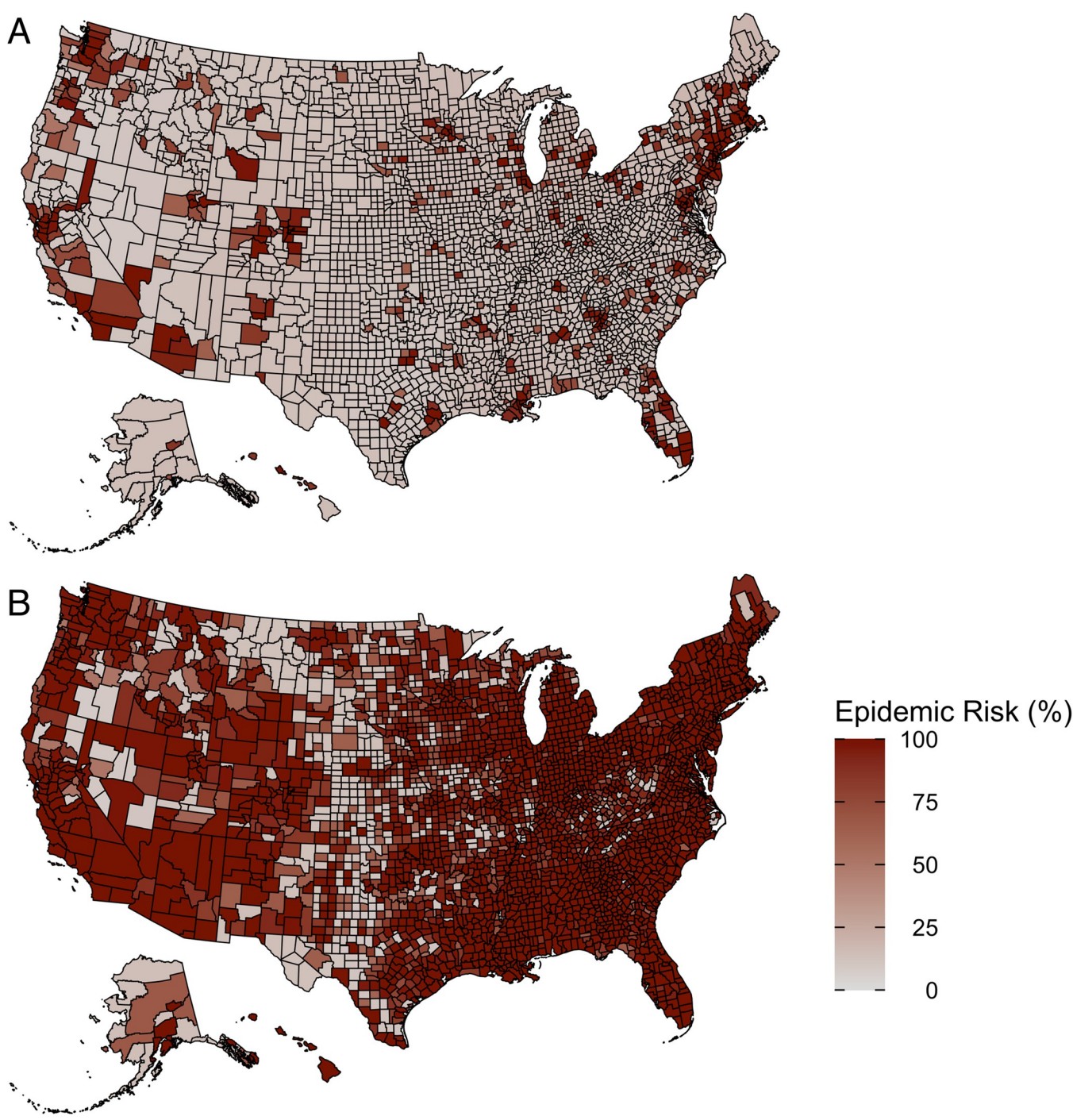

**Fig 2.** Estimated COVID-19 epidemic risk in 3,142 US counties as of March 16 (A) and April 13, 2020 (B). Epidemic risk was determined for each county based on its effective reproduction number ($R_e$) as estimated in [7], alongside the number of reported cases in the county on the specific date as described in Fig 1. Epidemic risk is the percent of 100,000 simulations for the county that become epidemics. We classified a simulation as an epidemic if it reached 2,000 cumulative infections and had a minimum prevalence of 50 new infections per day [5]. County-specific $R_e$ ranged from 1.4 to 4.4 with a median of 2.8. The model assumed a 10% case detection rate, generation time of 6.0 days, a latent period of 2.9 days, and infectious period of 6.2 days (Table 1 - retrospective).

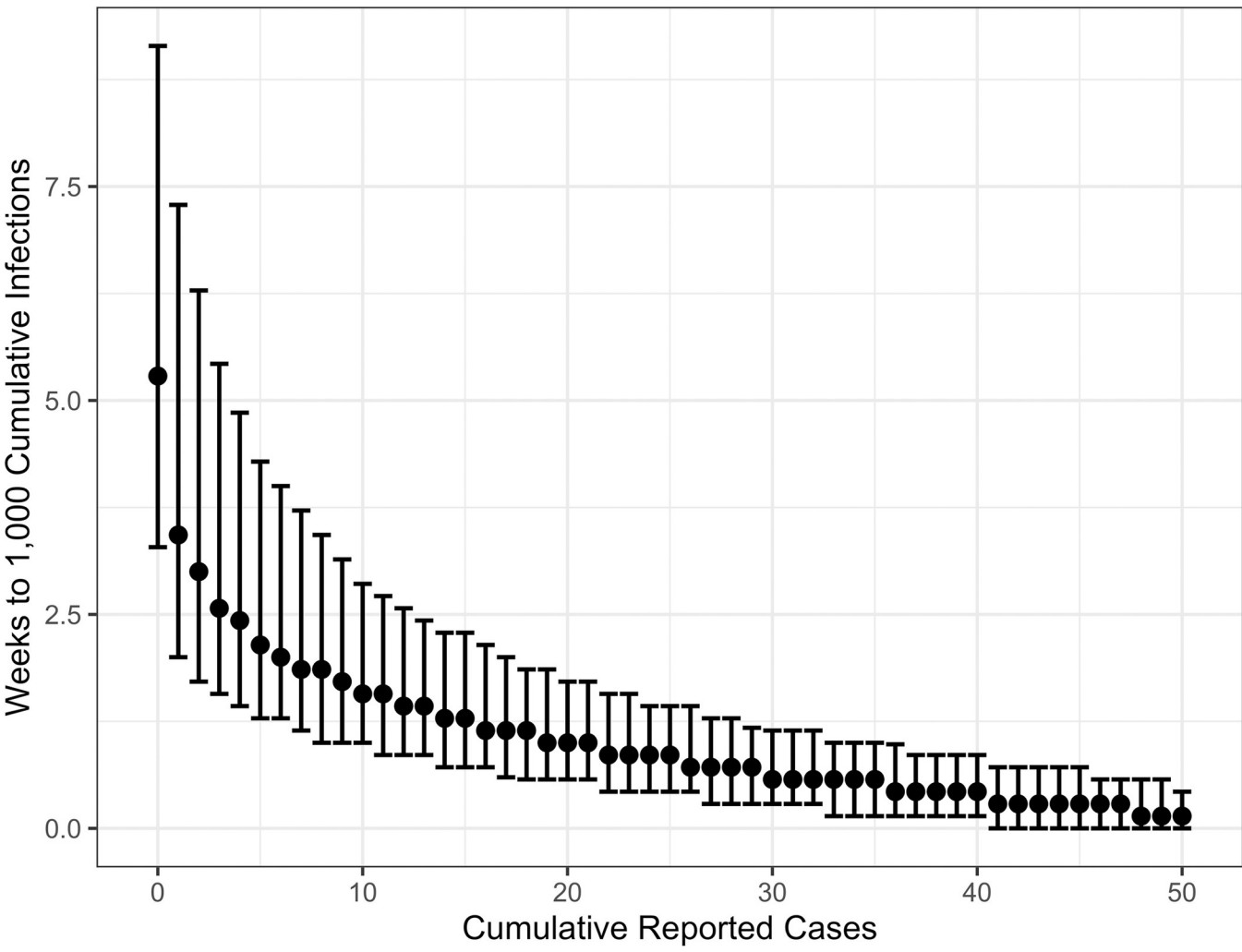

**Fig 3. Expected time until the local epidemic exceeds 1,000 cumulative infections in a county, assuming $R_e$ = 2.8, a 10% case detection rate, and generation time of 6.0 days.** For a given number of cumulative *reported cases* (x-axis), we assume an epidemic is underway then estimate the median and 95% CI (error bars) number of weeks until the cumulative *infections* reach or exceed 1,000. When the first case is reported, we expect cumulative infections to surpass 1,000 in 3.4 (95% CI 2.0–7.3) weeks; when the 10th case is reported, the expected lead time shrinks to 1.5 (95% CI 1.0–2.9) weeks. The estimates are based on 100,000 stochastic simulations of the retrospective model (Table 1).

US counties containing 94% of the US population had reported at least one COVID-19 case, resulting in over a 50% chance of having an epidemic (epidemic risk). We thus believe that our estimates may have helped communities understand the silent but rapid expansion of the virus. By that date, ten states (Alabama, Arkansas, Iowa, Missouri, Nebraska, North Dakota, South Carolina, South Dakota, Utah, and Wyoming) had not yet enacted statewide stay-at-home orders [20]. While our estimates may not have directly swayed state policy makers, they provided evidence in support of strong mitigation despite low reported case counts in many areas that took action.

As validation, we compared our epidemic risk estimates to the proportion of counties that reported additional cases between March 16 and March 23, 2020 (Fig 4 and S8 Fig). We found that our epidemic risk estimates significantly correlate with subsequent case increases. We limit our historical comparison to the period prior to March 23, 2020, after which

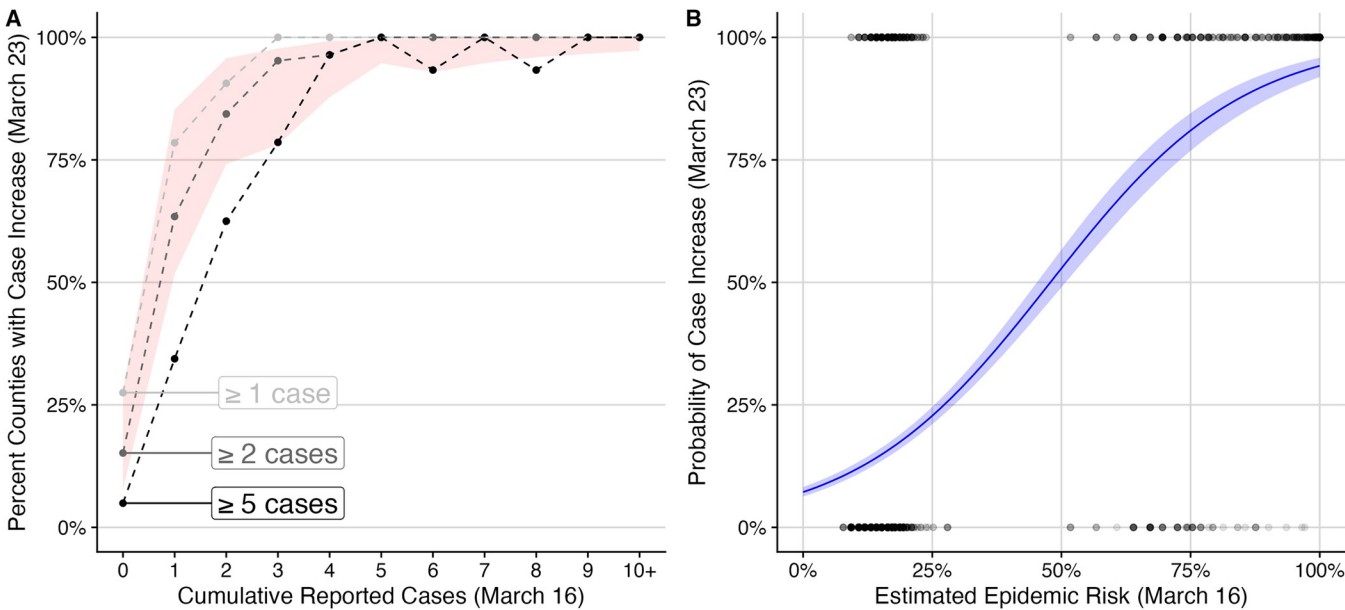

**Fig 4. Comparison of estimated epidemic risks and reported increases in cases at the county level between March 16 and March 23, 2020.** (A) Proportion of all US counties that had specified one-week increase in reported COVID-19 cases, compared to the cumulative case count in the county as of March 16, 2020 (x-axis) [16]. The light, medium and dark gray lines correspond to increases of at least one, two, or five new reported cases within one week, respectively. The red ribbon indicates model estimates for the probability that an epidemic is underway, depending on the cumulative reported cases. The bottom and top of the ribbon correspond to estimates for the lowest and highest risk counties across the United States, where risk is estimated based on county-specific estimates of $R_e$ and the cumulative number of reported cases on March 16. These estimates are calculated based on 100,000 simulations for each reproduction number ($R_e$ = 1.4 to 4.4 by 0.1), assuming a 10% case detection rate and a generation time of 6.0 days. (B) Estimates of epidemic risks on March 16 correlate with case count increases in the subsequent week across all counties. Points indicate whether counties reported at least two new COVID-19 cases between March 16 and March 23, where the bottom and top of the graph correspond to counties that did or did not report such increases. The line and shading indicate the estimated mean (line) and 95% confidence interval (ribbon) resulting from a logistic regression relating actual one-week reported increase to estimated risk on March 16, 2020. We estimate that a 10% increase in estimated risk corresponds to a 0.53 (95% CI: 0.49–0.58) increase in the log odds that the county reported at least two additional cases in the following week.

unprecedented COVID-19 lock downs and social distancing policies decreased epidemic risks [13, 21–24]. We also compare the results of our analysis with estimates that we originally made on March 16, 2020, before we had data that allowed us to estimate county-specific SARS-CoV-2 reproduction numbers. At that time, we made the simplifying assumption that transmission rates were uniform across counties. Our original estimates are consistent with both our retrospective estimates, though slightly less accurate (Fig 5). Importantly, even the data limited analyses provided clear indication of the extent of undetected epidemic risk and urgency of action across the US.

Our results are consistent with the current understanding of early COVID-19 transmission in the United States. Epidemiological and phylodynamic models identified substantial, undocumented, COVID-19 transmission leading up to stay-at-home orders in late March 2020 [14, 25], with non-pharmaceutical interventions reducing transmission and preventing infections and mortality [24, 26]. Proactive responses to COVID-19 have been estimated to shorten the duration of costly measures [27, 28], whereas delays have likely cost lives [26, 29]. Thus, our results suggest that the first reported case should trigger action if the goal is to fully contain an emerging outbreak as quickly as possible. The risk of an ongoing epidemic likely already exceeds 50% and delaying action may substantially reduce the window for corrective action and amassing adequate healthcare and other mitigation resources (Fig 3).

Our analyses make several key assumptions. First, case detection rates may vary geographically and change through time depending on testing availability and regulations. Our

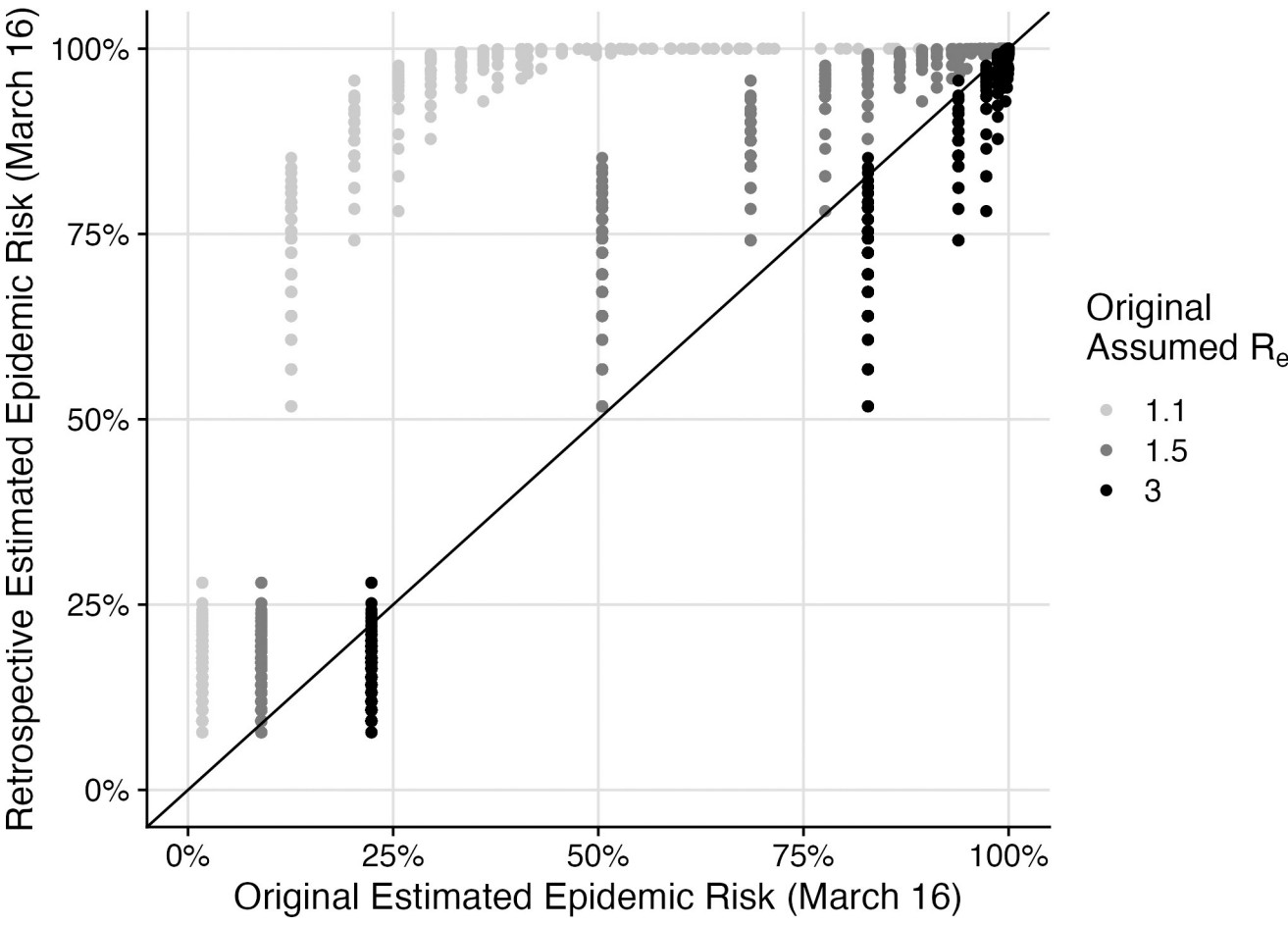

**Fig 5. Comparison of original epidemic risk estimates, assuming a uniform $R_e$ across counties, and retrospective estimates, assuming empirical county-level estimates of $R_e$ on March 16, 2020 across 3,142 US counties.** Each point corresponds to a pair of risk estimates (original on x-axis vs retrospective on y-axis) for a single county. Points are shaded according to the assumed effective reproduction number for the original estimate. The solid diagonal line indicates matching estimates.

assumption of 10% is based on a CDC seroprevalence study, which reported that rates ranged from 4% to 16% across ten sites [4]. Second, we modeled superspreading events based on estimates for SARS-CoV in Singapore in 2003 [15], which are consistent with more recent reports for SARS-CoV-2 [30–32]. Third, our estimates do not account for repeated importations of infected individuals, all simulations start with only one infected individual. Multiple importations would reduce our estimated levels of epidemic risk since reported cases could reflect independent clusters rather than continuous chains of transmission. Finally, we considered scenarios with lower effective reproduction numbers than estimated (Fig 1 and S3–S7 Figs), which may be more appropriate for the epidemic risks following the enactment of non-pharmaceutical interventions, but we did not account for changes to the effective reproduction number over time. While transmission can vary temporally depending on local policies, testing efforts [33, 34], and behavior [13], we made these simplifying assumption, because our overall goal was to estimate county-level epidemic risk in the absence of interventions.

While simple, this modeling framework provided useful insight during a highly uncertain time during the US Zika Virus epidemic in 2016 [5]. Now, we have shown how it can be quickly adapted to provide rapid situational awareness for COVID-19 in real-time, with

different epidemiological characteristics. Similar results between our original and retrospective analyses further validate the robustness and reliability of the original epidemiological risk estimates and provide the framework for implementation during future emerging infectious outbreaks. Overall, we find that for silently spreading pathogens, proactive control measures may be prudent, even before the threat becomes apparent [35].

## Methods

### Data

We obtained daily county-level estimates of confirmed and suspected COVID-19 cases from a data repository curated by the New York Times [16]. The US county map was based on TIGER/Line shapefiles provided by the US Census Bureau [36] and accessed through 'tidycensus' version 1.2.3 for the year 2019 [37]. Estimates of each county's 2019 population from the US Census Bureau [38] were used only to estimate the proportion of the population likely experiencing an epidemic (epidemic risk greater than 50%). For the original analysis, epidemic risk is based on a county's cumulative reported COVID-19 cases and the effective reproduction number ($R_e$) which is assumed to be the same for all counties. Our baseline scenario assumed the $R_e$ of SARS-CoV-2 was 1.5, accounting for ongoing social distancing measures across the US by mid-April, 2020 [39] and that 10% of all cases being reported [4]. Parameter estimates for the original analysis were taken from the literature available by mid-March, 2020 when much about SARS-CoV-2 was still unknown (Table 1).

For the retrospective analysis, epidemic risk is based on the county-specific cumulative reported cases and county-specific effective reproduction numbers ($R_e$). Epidemiological parameters for the model are drawn from a literature search carried out in December 2022, which updated the best estimate for the COVID-19 latent period from 1.25 to 2.9 days (see comparison in Table 1). We assume that the county $R_e$ equals the basic reproduction number estimated in [7] for all counties in the contiguous US. As population density and urban-rural classification are strong predictors for the COVID-19 reproduction number [9, 40], we estimated the $R_e$ for counties in Alaska and Hawaii as the mean $R_e$ of all the contiguous US counties with the same urban-rural designation code as defined by 2013 estimates from the National Center for Health Statistics Urban-Rural Classification Scheme for Counties rounded to the nearest tenth [10]. In total, counties had twenty-nine different $R_e$ values ranging from 1.4 to 4.4. We included $R_e$ = 1.1 as well to simulate a possible social distancing scenario when counties were under shelter-in-place orders [26].

### Model

We adapted the framework of another silent spreader–Zika Virus–which threatened to emerge in southern US states in 2016 [5] to model COVID-19 spread in US counties. The discrete-time SEIR model assumed a branching process for early transmission in which the number of secondary infections per infected case was distributed according to a negative binomial distribution to capture occasional superspreading events, as estimated for SARS-CoV outbreaks in 2003 [15]. Similar to the methods in [5], the exposure and infectious periods consisted of "boxcars", smaller consecutive compartments that each individual must pass through. Boxcars enforce the minimum number of days spent in each compartment and more accurately reflect the waiting time distribution of a negative binomial distribution [41, 42]. For example, an infectious period of 9.5 days could be modeled as one compartment with a daily transition rate of 1/9.5 or broken up into seven boxcars with a daily transition rate of 7/9.5.

We account for imperfect detection and COVID-19 specific epidemiological characteristics for both original and retrospective scenarios (Table 1). We did not explicitly model

asymptomatic or pre-symptomatic transmission and thus maintained a low detection probability for all infections in both scenarios. To assess the impact of parameter assumptions on our estimates of epidemic risk alongside the impact of behavioral and policy changes that might have altered the effective reproduction number, we conducted a sensitivity analysis that varied $R_e$ from 1.1 to 3.0 (S3–S5 Figs) and detection rates from 5% to 40% (S1 Fig).

Our goal was to estimate the probability that an outbreak in a region would become an epidemic based on the number of observed reported cases in the region and assuming no behavioral changes or public health interventions. As such, we ran 100,000 stochastic outbreak simulations per scenario ($R_e$ held constant) beginning with a single undetected case and ending when cumulative infections reached 2,000 or the outbreak died out (whichever came first). Because we modeled transmission as a branching process, the susceptible population did not deplete as in other compartmental SEIR models. Following the methodology of [5], simulated outbreaks that reached 2,000 cumulative infections and had a minimum prevalence of 50 new infections per day were classified as epidemics. Epidemic criteria were chosen conservatively to give self-limiting outbreaks sufficient time to die out and be differentiated from self-sustaining transmission chains (S10 Fig). If simulations were terminated too soon, then some self-limiting transmission chains may reach the maximum cumulative infections by chance. We calculated epidemic risk for a given number of detected cases, $x$, by looking at all outbreak simulations that had $x$ reported cases and calculating the proportion of those outbreaks that progressed to epidemics. For example, if 40,000 simulations satisfied the epidemic criteria, then the risk of an epidemic was 40%. For simulations that became epidemics (satisfied above epidemic criteria), we calculated the distribution of lags (in weeks) between the day the $x$th case was reported and the day the epidemic surpassed 1,000 cumulative infections (Fig 3 and S7 Fig). Confidence intervals were calculated with the quantile function in R version 3.6.1 for original and retrospective scenarios [43].

## County epidemic risk assignment

We matched county cumulative reported cases (confirmed and suspected from [16]) with epidemic risk for each US county based on simulated cumulative reported cases (original) and matching the county-specific $R_e$ (retrospective). For example, if a county reported ten cumulative cases on March 16, 2020 and had an estimated $R_e$ of 2.0, then epidemic risk is assigned as a look-up of simulations which detected ten cumulative cases under a constant $R_e$ of 2.0. The retrospective analysis used county-specific estimated $R_e$ as described above and cumulative cases. In the original analysis presented in the supplement, all counties have the same effective reproduction number of 1.1, 1.5, or 3.0, and only vary due to cumulative reported cases.

## Model validation

To validate the estimates of epidemic risk, we used the county-specific estimated epidemic risk on March 16, 2020 (x-axis) as a predictor for if US counties reported at least one, two, or five new cases over the week of March 16 to 23, 2020 (y-axis) in a logistic regression model. First, we calculated county-specific epidemic risk on March 16 as described above. Second, March 23rd case counts were subtracted from those on March 16 and the difference was classified as an increase of at least one, two, or five cases (three separate binary classifications). Finally, a logistic regression was fit to each classification independently to determine if the number of cases on March 16 was a significant predictor of new reported cases one week later. We compared case counts from Monday to Monday to avoid weekend reporting bias, and this week in mid-March was before most lockdowns took place in the US and saw only a moderate increase in daily tests nationally (from 20,000 to 60,000) [44]. We estimate logistic regression models

based on the retrospective analysis with county-specific transmission rates (Fig 4B and S2 Fig) and for the original analysis across all assumed effective reproduction numbers (S9 Fig). We compare the two analyses through the Pearson's product-moment correlation in estimated epidemic risks (Fig 5), the estimated logistic regression coefficients, and the model AIC and weights as calculated in the 'bbmle' R package version 1.0.25 [45]. As a secondary qualitative validation, we group counties by their cumulative reported case counts up to March 16, 2020 and estimate the proportion of those counties that reported one, two, or five new cases in the subsequent week. We then compare that estimated proportion with the range of epidemic risks estimated across those counties from March 16 (Fig 4A and S8 Fig).

## Supporting information

**S1 Fig. Sensitivity analysis of original March 16, 2020 risk estimates (assuming constant $R_e$) with respect to the assumed reproduction number ($R_e$) and case detection probability.** Percentage of US counties (left) or US population living in counties (right) that have greater than a 50% risk for sustained local transmission across varying assumed transmission rates (shade) and case detection probabilities (x-axis).
(TIF)

**S2 Fig. Comparison of estimated epidemic risks (based on data available as of March 16, 2020) and reported increases in cases at the county level between March 16 and March 23, 2020.** Points indicate the binary outcome for each county of whether it reported at least one (A) or five (B) new COVID-19 cases between March 16 and March 23. The bottom and top of the graph correspond to counties that did or did report such increases. The line and shading indicate the estimated mean (line) and 95% confidence interval (ribbon) resulting from a logistic regression relating actual one-week reported increase to estimated risk on March 16, 2020. We estimate that a 10% increase in model estimated epidemic risk for March 16 yields a 0.55 (95% CI 0.49–0.61) and 0.57 (95% CI 0.53–0.61) increase in the log odds that the county reported at least one or five additional cases in the following week, respectively.
(TIF)

**S3 Fig.** Original county-level estimates of ongoing COVID-19 epidemics assuming $R_e$ = 1.5 for (A) March 16, 2020 and (B) April 13, 2020. Estimated epidemic risk increased from 9% for zero cases to 50% when one case was detected and 100% for twenty-five or more cases. (A) By March 16, 2020, epidemic risk exceeded 50% in roughly 15% of the 3,142 counties covering 63% of the US population. (B) By April 13, 2020, we estimated that over 85% of US counties comprising 96% of the national population had at least a 50% chance of having an epidemic already underway. The estimates assume a 10% case detection rate and generation time of 6.0 days.
(TIF)

**S4 Fig.** Original county-level estimates of ongoing COVID-19 epidemics assuming $R_e$ = 1.1 for (A) March 16, 2020 and (B) April 13, 2020. Epidemic risk increased from 2% for zero cases to 13% when one case was detected. An $R_e$ of 1.1 may be appropriate for counties with strict social distancing measures. The model assumes the original parameter estimates, including a 10% case detection rate and generation time of 6.0 days.
(TIF)

**S5 Fig.** Original county-level estimates of ongoing COVID-19 epidemics assuming $R_e$ = 3.0 for (A) March 16, 2020 and (B) April 13, 2020. Epidemic risk increased from 22% for zero cases to 83% when one case was detected. The model also assumes the original parameter

estimates, including a 10% case detection rate and generation time of 6.0 days.
(TIF)

**S6 Fig. Sensitivity analysis of original estimates (assuming uniform $R_e$ across counties) with respect to the effective reproduction number ($R_e$).** For a given number of reported cases, the estimated risk of an epidemic increased with $R_e$. By the time a single case is reported, there is a 13%, 50% or 83% chance of an ongoing epidemic for an $R_e$ of 1.1, 1.5 or 3.0, respectively.
(TIF)

**S7 Fig.** Expected time until epidemic exceeds 1,000 cumulative infections in a county, assuming (A) Re = 1.5 and (B) 3.0, a 10% case detection rate, and generation time of 6.0 days. For a given number of cumulative reported cases (x-axis), we estimate the median and 95% CI (error bars) number of weeks until the cumulative infections reach or exceed 1,000 for simulations classified as epidemics. (A) For Re = 1.5, when the first case is reported cumulative infections surpass 1,000 in 7.5 (95% CI 3.9–16.3) weeks; when the 10th case is reported, the expected lead time shrinks to 4.4 (95% CI 2.1–11.4) weeks. (B) Increasing Re to 3.0 there is a lag time of 3.0 (95% CI 1.6–6.6) weeks for the first case that decreases to 1.1 (95% CI 0.6–2.4) by the tenth case. Negative estimates suggest that 1,000 infections are reached prior to reporting a certain number of cumulative cases. The estimates are based on 100,000 stochastic simulations per Re.
(TIF)

**S8 Fig. Comparison between original estimates of epidemic risk on March 16, 2020 (assuming constant $R_e$ across counties) and observed percent of US counties in which COVID-19 case counts increased from March 16 to March 23, 2020, as a function of cumulative reported cases as of March 16, 2020.** The light, medium and dark gray lines correspond to increases of at least one, two, or five new reported cases within one week, respectively. The red ribbon indicates the original model estimated epidemic risk, given the cumulative reported cases on March 16, 2020 indicated on the x-axis. The bottom and top of the ribbon correspond to estimates assuming $R_e = 1.5$ and $R_e = 3.0$, respectively. These estimates are calculated based on 100,000 simulations for each reproduction number, assuming a 10% case detection rate and a generation time of 6.0 days. The odds of a county detecting at least five new cases increased by 4.90 (95% CI 4.14–5.99) for every one unit increase in cases on March 16. For example, a county reporting only one case as of March 16 was roughly five times more likely to report at least six new cases a week later than a county with no previously reported cases.
(TIF)

**S9 Fig. Correlations between original epidemic risk estimates for March 16, 2020 (assuming uniform $R_e$) and case count increases in the subsequent week across all counties.** Points indicate the binary outcome for each county of whether it reported at least one, two, or five (rows) new COVID-19 cases between March 16 and March 23 under different assumed effective reproduction numbers (columns). Lines and ribbons indicate the estimated means and 95% confidence intervals for the fitted logistic regression models.
(TIF)

**S10 Fig. Outbreak simulation length determined by $R_e$ close to one with both original and retrospective parameters.** If simulations end too early, then more than expected can reach sufficient cumulative infections by random chance alone and not reflect true epidemics. By ending simulations at 2,000 cumulative infections we could confidently separate simulations with $R_e$ just below one (0.95, top row) from the epidemics of those with $R_e$ just above one

(1.05, bottom row). For $R_e$ = 0.95 0.0% of original and 0.003% of retrospective simulations reached 2,000 cumulative infections and met a minimum prevalence of 50 new infections. As $R_e$ increased to 1.05, 0.11% of original and 0.10% of retrospective simulations were classified as epidemics. Ending simulations at 2,000 cumulative infections and requiring a minimum prevalence of 50 new infections per day is sufficient to distinguish self-limiting simulations from self-sustaining.
(TIF)

## Author Contributions

**Conceptualization:** Emily M. Javan, Spencer J. Fox, Lauren Ancel Meyers.

**Data curation:** Emily M. Javan.

**Formal analysis:** Emily M. Javan, Spencer J. Fox.

**Funding acquisition:** Lauren Ancel Meyers.

**Investigation:** Emily M. Javan, Spencer J. Fox.

**Methodology:** Emily M. Javan, Spencer J. Fox.

**Supervision:** Lauren Ancel Meyers.

**Validation:** Emily M. Javan, Spencer J. Fox.

**Visualization:** Emily M. Javan, Spencer J. Fox.

**Writing – original draft:** Emily M. Javan, Spencer J. Fox.

**Writing – review & editing:** Emily M. Javan, Spencer J. Fox, Lauren Ancel Meyers.

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
