## [Decision Letter · Decision Letter 0]

7 Oct 2022

PONE-D-22-17433Estimating the unseen emergence of COVID-19 in the US

PLOS ONE

Dear Dr. Javan,

Thank you for submitting your manuscript to PLOS ONE. After careful consideration, we feel that it has merit but does not fully meet PLOS ONE’s publication criteria as it currently stands. Therefore, we invite you to submit a revised version of the manuscript that addresses the points raised during the review process.

We look forward to receiving your revised manuscript.

Kind regards,

Sung-mok Jung

Academic Editor

PLOS ONE

Journal Requirements

Additional Editor Comments:

I agree with the authors that early risk assessment is essential, and the model developed for the Zika virus can aid in determining the probability of undetected transmissions. However, as both reviewers mentioned, I also have concerns on the model validation and assumptions used in the model (including parameters). Thus, I would be great if authors could response to those comments and add additional analyses to validate the proposed model by comparing model estimates with the empirically observed data.

Reviewers' comments:

Reviewer's Responses to Questions

**Comments to the Author**

1. Is the manuscript technically sound, and do the data support the conclusions?

Reviewer #1: Yes

Reviewer #2: Partly

2. Has the statistical analysis been performed appropriately and rigorously? 

Reviewer #1: Yes

Reviewer #2: No

3. Have the authors made all data underlying the findings in their manuscript fully available?

Reviewer #1: Yes

Reviewer #2: Yes

4. Is the manuscript presented in an intelligible fashion and written in standard English?

Reviewer #1: Yes

Reviewer #2: Yes

5. Review Comments to the Author

Reviewer #1: The authors present results from an SEIR model developed for Zika that was adapted to SARS-CoV-2 to determine risk of an undetected outbreak by county given x number of reported cases in that county.

The author summary suggests that the model was developed to support local decision makers during the early phase of the pandemic, but there is no discussion of this point. I think that if the authors are planning to frame their article as developed to support local decision-making, they need to spend a substantially more amount of text discussing/analyzing this. Was the model used? Was any validation of the model done? How long did it take for counties to reach 1,000 cumulative cases (Fig 3)?

How do the authors define an “epidemic” for their purpose of evaluating epidemic risk? (Especially with regard to interpretation of Figure 1)

While I appreciate the value of a simple model, I do believe the framing of the paper requires more evaluation/discussion of the practical implications of the model. If the intent of the authors is not to do that, it would be ideal to reframe the article to emphasize potential for future use of the model rather than utility of the model during the early phase of the COVID-19 pandemic (or some other reframing).

Minor notes:

“Undetected” epidemic is a phrase more commonly used to describe what the authors assess rather than “unseen” epidemic, which has more of a literary feel.

The description of COVID-19 “waves” in the first paragraph of the results seems a non sequitur from the introduction as the focus of this is the earliest phase of the pandemic before it was truly grasped that there would be so many waves.

Line 48: the long-form name for COVID-19 is “coronavirus disease 2019”—if the authors wish/prefer to use “novel coronavirus” as a nod to the original name for the disease, it would be prudent to do so separately from establishment of the acronym.

Line 179: Please define/explain the use of the term boxcars.

Fig 3: Please define total cases and how this is different from reported cases.

Reviewer #2: In this manuscript the authors repurposed a model that was previously used to study the spread of the Zika Virus to estimate the probability of unseen transmission. The model uses a stochastic SEIR branching process to simulate the outbreak until April 2020. The authors find that counties that have confirmed one case have a 50% probability of an epidemic. The authors did a lot of work reconstructing the early stage of the epidemic for each county in the US. Understanding the early phase of an emerging epidemic is crucial and I applaud and thank the authors for their hard work providing situational awareness to both public health departments and the public.

With the current version of the paper I have a few issues and concerns about the model assumptions and I believe there is a lack of validation. I have broken down my comments into Major and Minor issues below.

Major issues

-Model assumptions: From my understanding of the paper, it says that all counties have the same Re which is either 1.1, 1.5, or 3 and that is held constant throughout the entire stochastic process. If this is the case, I do not believe this is appropriate. In addition, I do not think it is true that all counties had similar transmission rates. I agree with the authors that in early March the Re was most likely closer to 3 due to the lack of interventions. However, it changed when places started social distancing/ working from home/ only opening essential services. There are multiple data sources out there reporting the Rt and changes in mobility (some at the county level) that could be incorporated to adjust for this. I know it is a lot of work but including this heterogeneity could change the results. I am also concerned about the estimate for the Latent period, this value (1.25 days) is one of the smaller values I have seen for the early estimates of this parameter. Most of the values I have read are between 2-7 days.

-Model calibration: Can the authors also confirm the calibration procedure for each individual county? From what I read, in order to get the variation in the timing of individual counties, stochastic runs are shifted to match the case data. However, I’m not entirely sure exactly how this is done. Do you choose one date (March 16) or a group of dates? I guess I’m not sure what the “fitting window” is and how the model is mapped to actual dates because the model doesn’t use information on importations, the timing of the start of the outbreak… etc.

-Validation: This paper is missing statistical validation of the model estimates vs. observational data. The model is currently validated by looking at whether case counts increased in the following week. In Figure 4, the model estimates are overlapped with the values using reported cases. However, I think a more quantitative way of validating the model against the surveillance data should be used. Possibly some correlation measure. Another type of validation measure that could be useful is based on the timing of community transmission at the county level (answering “when” did counties have widespread transmission and was it similar to what we observed with reported cases). So looking at the temporal evolution of the early phase. I also think that this paper is missing a discussion on how this analysis aligns with the literature. There has been a lot of work on the early phase of the epidemic using phylogenetic analysis, retrospective samples, waste water surveillance, and other epidemic modeling approaches within the US (not much at the county level). While there isn’t a single, comprehensive paper to compare this work to, it would contribute to the validation of the model and deepen the discussion.

Minor Issue:

-Language of unseen transmission: As of now, the paper uses many different words and phrases to describe an “unseen epidemic” and how to estimate it such as: epidemic risk, unseen transmission, widespread transmission, sustained community transmission, and outbreaks that spread widely. It would useful and helpful to the reader to use a common terminology that is defined explicitly in the main paper. From what I read “epidemic risk” is the most well defined, but I’ll leave that up to the authors to decide what works best.

6. PLOS authors have the option to publish the peer review history of their article (what does this mean?). If published, this will include your full peer review and any attached files.

Reviewer #1: No

Reviewer #2: No

---

## [Author Response · Author response to Decision Letter 0]

7 Feb 2023

Please find our response to reviewers in the attached file, but I have copied the text without figures here as well.

Response to Reviewers: 

“Estimating the undetected emergence of COVID-19 in the US”

Emily M. Javan, Spencer J. Fox, Lauren Ancel Meyers

Editor

 I agree with the authors that early risk assessment is essential, and the model developed for the Zika virus can aid in determining the probability of undetected transmissions. However, as both reviewers mentioned, I also have concerns on the model validation and assumptions used in the model (including parameters). Thus, it would be great if authors could respond to those comments and add additional analyses to validate the proposed model by comparing model estimates with the empirically observed data.

 We thank the editor and the reviewers for the thorough comments regarding our manuscript. As described below and in response to the feedback we have now (1) revised our epidemic risk model to incorporate recent parameter estimates and spatial heterogeneity in transmission risk, (2) performed further model validation using county case increases and epidemic risk estimates from both our original and retrospective models, (3) expanded our discussion of the utility of the study, and (4) clarified the methodology.

 Map license confirmation

 The TIGER/Lines shapefiles used for mapping US counties were produced by the US Census Bureau and do not have any copyright protection (Title 17 U.S.C., Section 105). We have added a citation to the methods section to acknowledge the US Census Bureau and the R package used to access the shapefiles: 

 “The US county map was based on TIGER/Line shapefiles provided by the US Census Bureau [36] and accessed through ‘tidycensus’ version 1.2.3 for the year 2019 [37].”

Reviewer #1 

The authors present results from an SEIR model developed for Zika that was adapted to SARS-CoV-2 to determine risk of an undetected outbreak by county given x number of reported cases in that county.

Major comments:

 The author summary suggests that the model was developed to support local decision makers during the early phase of the pandemic, but there is no discussion of this point. I think that if the authors are planning to frame their article as developed to support local decision-making, they need to spend a substantially more amount of text discussing/analyzing this. A) Was the model used? B) Was any validation of the model done? C) How long did it take for counties to reach 1,000 cumulative cases (Fig 3)?

 A) We now address these issues in the Discussion as follows:

 “Our framework was developed in the first months of the COVID-19 pandemic to provide local situational awareness for both government officials and the public, at a time when the data were highly uncertain. On April 3, 2020, the New York Times published our first set of estimates as a national risk map appearing on the front page [18], which reached an estimated 694 million unique viewers according to Meltwater [19]. The map estimated that 70% of 3,142 US counties containing 94% of the US population had reported at least one COVID-19 case, resulting in over a 50% chance of having an epidemic (epidemic risk). We thus believe that our estimates may have helped communities understand the silent but rapid expansion of the virus. By that date, ten states (Alabama, Arkansas, Iowa, Missouri, Nebraska, North Dakota, South Carolina, South Dakota, Utah, and Wyoming) had not yet enacted statewide stay-at-home orders [20]. While our estimates may not have directly swayed state policy makers, they provided evidence in support of strong mitigation despite low reported case counts in many areas that took action.”

 B) As additional validation, we now compare our model estimates of epidemic risk to county case increases from March 16 to March 23, 2020, and have made the following updates: 

 Added Fig 4B and S2 which compare estimated epidemic risk on March 16 to the probability of case increases by one (S2A), two (2B), or five (S2B) in a county by March 23, 2020 with logistic regression. 

Fig 4. Comparison of estimated epidemic risks and reported increases in cases at the county level between March 16 and March 23, 2020. (A) Proportion of all US counties that had specified one-week increase in reported COVID-19 cases, compared to the cumulative case count in the county as of March 16, 2020 (x-axis) [16]. The light, medium and dark gray lines correspond to increases of at least one, two, or five new reported cases within one week, respectively. The red ribbon indicates model estimates for the probability that an epidemic is underway, depending on the cumulative reported cases. The bottom and top of the ribbon correspond to estimates for the lowest and highest risk counties across the United States, where risk is estimated based on county-specific estimates of Re and the cumulative number of reported cases on March 16. These estimates are calculated based on 100,000 simulations for each reproduction number (Re=1.4 to 4.4 by 0.1), assuming a 10% case detection rate and a generation time of 6.0 days. (B) Estimates of epidemic risks on March 16 correlate with case count increases in the subsequent week across all counties. Points indicate whether counties reported at least two new COVID-19 cases between March 16 and March 23, where the bottom and top of the graph correspond to counties that did or did report such increases. The line and shading indicate the estimated mean (line) and 95% confidence interval (ribbon) resulting from a logistic regression relating actual one-week reported increase to estimated risk on March 16, 2020. We estimate that a 10% increase in estimated risk corresponds to a 0.53 (95% CI: 0.49-0.58) increase in the log odds that the county reported at least two additional cases in the following week.

S2 Fig. Comparison of estimated epidemic risks (based on data available as of March 16, 2020) and reported increases in cases at the county level between March 16 and March 23, 2020. Points indicate the binary outcome for each county of whether it reported at least one (A) or five (B) new COVID-19 cases between March 16 and March 23. The bottom and top of the graph correspond to counties that did or did report such increases. The line and shading indicate the estimated mean (line) and 95% confidence interval (ribbon) resulting from a logistic regression relating actual one-week reported increase to estimated risk on March 16, 2020. We estimate that a 10% increase in model estimated epidemic risk for March 16 yields a 0.55 (95% CI 0.49-0.61) and 0.57 (95% CI 0.53-0.61) increase in the log odds that the county reported at least one or five additional cases in the following week, respectively.

 Updated the results as follows: “To validate our model, we compare the estimated epidemic risk on March 16, 2020 to increases in reported case counts in the following week (Fig 4A). We find that our estimates of epidemic risk correlate significantly with the probability that a county reported additional cases in the subsequent week (logistic regression, p<0.001). A 10% increase in estimated risk corresponds to an increase in the log odds of a county detecting at least one, two, or five new cases of 0.55 (95% CI 0.49-0.61), 0.53 (95% CI 0.49-0.58), and 0.57 (95% CI 0.53-0.61), respectively (Fig 4B and S2 Fig).”

 The new Figure 5 and Figures S1, S3-S9 serve as model validation and sensitivity analyses of the original (April 4, 2020) parameterization of the model. Figure 5 compares risk estimates based on retrospective county-level estimates of Re to our original estimates assuming a uniform Re across all counties, see Table 1 for comparison of parameters. The retrospective estimates most closely match our original estimates assuming an Re=3.0. Figures S3-S5 map our original risk estimates assuming uniform Re across counties. Figures S1 and S6 are sensitivity analyses of the original model demonstrating the effect on epidemic risk when we vary the case detection probability and Re, respectively. Figure 3 and S7 show the time (weeks) to 1,000 cumulative infections in simulations that became epidemics given a number of reported cases. As Re or the number of detected cases increases the lead time to respond to the growing epidemic shrinks. Figures S8 and S9 are the uniform Re versions of Figure 4A and 4B model validation.

Fig 5. Comparison of original epidemic risk estimates, assuming a uniform Re across counties, and retrospective estimates, assuming empirical county-level estimates of Re on March 16, 2020 across 3,142 US counties. Each point corresponds to a pair of risk estimates (original on x-axis vs retrospective on y-axis) for a single county. Points are shaded according to the assumed effective reproduction number for the original estimate. The solid diagonal line indicates matching estimates.

S1 Fig. Sensitivity analysis of original March 16, 2020 risk estimates (assuming constant Re) with respect to the assumed reproduction number (Re) and case detection probability. Percentage of US counties (left) or US population living in counties (right) that have greater than a 50% risk for sustained local transmission across varying assumed transmission rates (shade) and case detection probabilities (x-axis).

S3 Fig. Original county-level estimates of ongoing COVID-19 epidemics assuming Re=1.5 for (A) March 16, 2020 and (B) April 13, 2020. Estimated epidemic risk increased from 9% for zero cases to 50% when one case was detected and 100% for twenty-five or more cases. (A) By March 16, 2020, epidemic risk exceeded 50% in roughly 15% of the 3,142 counties covering 63% of the US population. (B) By April 13, 2020, we estimated that over 85% of US counties comprising 96% of the national population had at least a 50% chance of having an epidemic already underway. The estimates assume a 10% case detection rate and generation time of 6.0 days. 

S4 Fig. Original county-level estimates of ongoing COVID-19 epidemics assuming Re=1.1 for (A) March 16, 2020 and (B) April 13, 2020. Epidemic risk increased from 2% for zero cases to 13% when one case was detected. An Re of 1.1 may be appropriate for counties with strict social distancing measures. The model assumes the original parameter estimates, including a 10% case detection rate and generation time of 6.0 days.

S5 Fig. Original county-level estimates of ongoing COVID-19 epidemics assuming Re=3.0 for (A) March 16, 2020 and (B) April 13, 2020. Epidemic risk increased from 22% for zero cases to 83% when one case was detected. The model also assumes the original parameter estimates, including a 10% case detection rate and generation time of 6.0 days.

S6 Fig. Sensitivity analysis of original estimates (assuming uniform Re across counties) with respect to the effective reproduction number (Re). For a given number of reported cases, the estimated risk of an epidemic increased with Re. By the time a single case is reported, there is a 13%, 50% or 83% chance of an ongoing epidemic for an Re of 1.1, 1.5 or 3.0, respectively. 

Fig 3. Expected time until the local epidemic exceeds 1,000 cumulative infections in a county, assuming Re=2.8, a 10% case detection rate, and generation time of 6.0 days. For a given number of cumulative reported cases (x-axis), we assume an epidemic is underway then estimate the median and 95% CI (error bars) number of weeks until the cumulative infections reach or exceed 1,000. When the first case is reported, we expect cumulative infections to surpass 1,000 in 3.4 (95% CI 2.0-7.3) weeks; when the 10th case is reported, the expected lead time shrinks to 1.5 (95% CI 1.0-2.9) weeks. The estimates are based on 100,000 stochastic simulations of the retrospective model (Table 1).

S7 Fig. Expected time until epidemic exceeds 1,000 cumulative infections in a county, assuming (A) Re=1.5 and (B) 3.0, a 10% case detection rate, and generation time of 6.0 days. For a given number of cumulative reported cases (x-axis), we estimate the median and 95% CI (error bars) number of weeks until the cumulative infections reach or exceed 1,000 for simulations classified as epidemics. (A) For Re=1.5, when the first case is reported cumulative infections surpass 1,000 in 7.5 (95% CI 3.9-16.3) weeks; when the 10th case is reported, the expected lead time shrinks to 4.4 (95% CI 2.1-11.4) weeks. (B) Increasing Re to 3.0 there is a lag time of 3.0 (95% CI 1.6-6.6) weeks for the first case that decreases to 1.1 (95% CI 0.6-2.4) by the tenth case. Negative estimates suggest that 1,000 infections are reached prior to reporting a certain number of cumulative cases. The estimates are based on 100,000 stochastic simulations per Re.

S8 Fig. Comparison between original estimates of epidemic risk on March 16, 2020 (assuming constant Re across counties) and observed percent of US counties in which COVID-19 case counts increased from March 16 to March 23, 2020, as a function of cumulative reported cases as of March 16, 2020. The light, medium and dark gray lines correspond to increases of at least one, two, or five new reported cases within one week, respectively. The red ribbon indicates the original model estimated epidemic risk, given the cumulative reported cases on March 16, 2020 indicated on the x-axis. The bottom and top of the ribbon correspond to estimates assuming Re=1.5 and Re=3.0, respectively. These estimates are calculated based on 100,000 simulations for each reproduction number, assuming a 10% case detection rate and a generation time of 6.0 days. The odds of a county detecting at least five new cases increased by 4.90 (95% CI 4.14-5.99) for every one unit increase in cases on March 16. For example, a county reporting only one case as of March 16 was roughly five times more likely to report at least six new cases a week later than a county with no previously reported cases. 

S9 Fig. Correlations between original epidemic risk estimates for March 16, 2020 (assuming uniform Re) and case count increases in the subsequent week across all counties. Points indicate the binary outcome for each county of whether it reported at least one, two, or five (rows) new COVID-19 cases between March 16 and March 23 under different assumed effective reproduction numbers (columns). Lines and ribbons indicate the estimated means and 95% confidence intervals for the fitted logistic regression models. 

 We describe these results in a new results section: “As additional validation of the modeling framework, we compare the estimates originally made in March 2020, before we had county-specific estimates of reproduction numbers (Table 1, Fig 5). At that time, we assumed all counties had the same effective reproduction numbers, ranging from 1.1 to 3.0; we also originally assumed a latent period of 1.25 rather than 2.9 days. Our original estimates assuming Re=3.0 most closely match the retrospective estimates (Pearson's product-moment correlation, r=0.99; p<0.001). Our original county-level risk maps (S3-S5 Figs) and estimates for the time until counties will reach 1,000 cumulative infections (S6 and S7 Figs) are also consistent with our retrospective analysis. Finally, our original estimates reliably predicted subsequent county case increases (S9 Fig). For example, assuming Re=3.0, a 10% increase in estimated epidemic risk corresponds to an increase in the log odds of a county detecting at least one, two, or five new cases by March 23 of 0.48 (95% CI: 0.43-0.53), 0.49 (95% CI: 0.45-0.53), and 0.55 (95% CI: 0.51-0.59), respectively.”

 We now discuss the validation in more depth in the discussion section: “As validation, we compared our epidemic risk estimates to the proportion of counties that reported additional cases between March 16 and March 23, 2020 (Fig 4 and S8). We found that our epidemic risk estimates significantly correlate with subsequent case increases. We limit our historical comparison to the period prior to March 23, 2020, after which unprecedented COVID-19 lock downs and social distancing policies decreased epidemic risks [13,21–24]. We also compare the results of our analysis with estimates that we originally made on March 16, 2020, before we had data that allowed us to estimate county-specific SARS-CoV-2 reproduction numbers. At that time, we made the simplifying assumption that transmission rates were uniform across counties. Our original estimates are consistent with both our retrospective estimates, though slightly less accurate (Fig 5). Importantly, even the data limited analyses provided clear indication of the extent of undetected epidemic risk and urgency of action across the US.”

 C) We cannot compare the estimates for the time to reach 1,000 cumulative infections, because these include both detected and undetected infections, and because non-pharmaceutical interventions dramatically changed the underlying transmission rates which we do not model. 

 We have now added this sentence to the discussion: “We limit our historical comparison to the period prior to March 23, 2020, after which unprecedented COVID-19 lock downs and social distancing policies decreased epidemic risks [13,21–24].” 

 We also justify these choices as part of the limitations section of our discussion: “Finally, we considered scenarios with lower effective reproduction numbers than estimated (Fig 1 and S3-S7 Figs), which may be more appropriate for the epidemic risks following the enactment of non-pharmaceutical interventions, but we did not account for changes to the effective reproduction number over time.”

 How do the authors define an “epidemic” for their purpose of evaluating epidemic risk? (Especially with regard to interpretation of Figure 1)

 We now included explicit definitions in the introduction and figure legends, and we have added more clarification in our methodology alongside justification of the choices in a new supplemental figure.

 Epidemic definition in introduction: “In this study, we use the term epidemic to refer to the county-level reproduction number of SARS-CoV-2 being greater than one. This is the threshold between self-sustained epidemic growth versus stuttering chains of transmission [6].”

 Epidemic definition for our simulation from Fig 2 legend: “Epidemic risk is the percent of 100,000 simulations for the county that become epidemics. We classified a simulation as an epidemic if it reached 2,000 cumulative infections and had a minimum prevalence of 50 new infections per day [5].” 

 New methodology text: “Our goal was to estimate the probability that an outbreak in a region would become an epidemic based on the number of observed reported cases in the region and assuming no behavioral changes or public health interventions. As such, we ran 100,000 stochastic outbreak simulations per scenario (Re held constant) beginning with a single undetected case and ending when cumulative infections reached 2,000 or the outbreak died out (whichever came first). Because we modeled transmission as a branching process, the susceptible population did not deplete as in other compartmental SEIR models. Following the methodology of [5], simulated outbreaks that reached 2,000 cumulative infections and had a minimum prevalence of 50 new infections per day were classified as epidemics. Epidemic criteria were chosen conservatively to give self-limiting outbreaks sufficient time to die out and be differentiated from self-sustaining transmission chains (S10 Fig). If simulations were terminated too soon, then some self-limiting transmission chains may reach the maximum cumulative infections by chance. We calculated epidemic risk for a given number of detected cases, x, by looking at all outbreak simulations that had x reported cases and calculating the proportion of those outbreaks that progressed to epidemics. For example, if 40,000 simulations satisfied the epidemic criteria, then the risk of an epidemic was 40%. For simulations that became epidemics (satisfied above epidemic criteria), we calculated the distribution of lags (in weeks) between the day the xth case was reported and the day the epidemic surpassed 1,000 cumulative infections (Fig 3 and S7 Fig). Confidence intervals were calculated with the quantile function in R version 3.6.1 for original and retrospective scenarios [43].”

 New supplemental figure:

S10 Fig. Outbreak simulation length determined by Re close to one with both original and retrospective parameters. If simulations end too early, then more than expected can reach sufficient cumulative infections by random chance alone and not reflect true epidemics. By ending simulations at 2,000 cumulative infections we could confidently separate simulations with Re just below one (0.95, top row) from the epidemics of those with Re just above one (1.05, bottom row). For Re=0.95 0.0% of original and 0.003% of retrospective simulations reached 2,000 cumulative infections and met a minimum prevalence of 50 new infections. As Re increased to 1.05, 0.11% of original and 0.10% of retrospective simulations were classified as epidemics. Ending simulations at 2,000 cumulative infections and requiring a minimum prevalence of 50 new infections per day is sufficient to distinguish self-limiting simulations from self-sustaining.

 While I appreciate the value of a simple model, I do believe the framing of the paper requires more evaluation/discussion of the practical implications of the model. If the intent of the authors is not to do that, it would be ideal to reframe the article to emphasize potential for future use of the model rather than utility of the model during the early phase of the COVID-19 pandemic (or some other reframing).

 We appreciate the helpful feedback, and now include more discussion of the utility of the framework both for COVID-19 and beyond.

 We have included new text in the discussion to frame the practical implications of the results from COVID-19: “Our framework was developed in the first months of the COVID-19 pandemic to provide local situational awareness for both government officials and the public, at a time when the data were highly uncertain. On April 3, 2020, the New York Times published our first set of estimates as a national risk map appearing on the front page [18], which reached an estimated 694 million unique viewers according to Meltwater [19]. The map estimated that 70% of 3,142 US counties containing 94% of the US population had reported at least one COVID-19 case, resulting in over a 50% chance of having an epidemic (epidemic risk). We thus believe that our estimates may have helped communities understand the silent but rapid expansion of the virus. By that date, ten states (Alabama, Arkansas, Iowa, Missouri, Nebraska, North Dakota, South Carolina, South Dakota, Utah, and Wyoming) had not yet enacted statewide stay-at-home orders [20]. While our estimates may not have directly swayed state policy makers, they provided evidence in support of strong mitigation despite low reported case counts in many areas that took action.”

 We also included new text to discuss the future usefulness of the modeling framework: “While simple, this modeling framework provided useful insight during a highly uncertain time during the US Zika Virus epidemic in 2016 [5]. Now, we have shown how it can be quickly adapted to provide rapid situational awareness for COVID-19 in real-time, with different epidemiological characteristics. Similar results between our original and retrospective analyses further validate the robustness and reliability of the original epidemiological risk estimates and provide the framework for implementation during future emerging infectious outbreaks. Overall, we find that for silently spreading pathogens, proactive control measures may be prudent, even before the threat becomes apparent [35].”

Minor comments:

 “Undetected” epidemic is a phrase more commonly used to describe what the authors assess rather than “unseen” epidemic, which has more of a literary feel.

 We have changed all instances of “unseen” to “undetected” throughout the manuscript.

 The description of COVID-19 “waves” in the first paragraph of the results seems a non sequitur from the introduction as the focus of this is the earliest phase of the pandemic before it was truly grasped that there would be so many waves.

 The term “waves” was changed to epidemic for clarity and consistency. 

 Line 48: the long-form name for COVID-19 is “coronavirus disease 2019”—if the authors wish/prefer to use “novel coronavirus” as a nod to the original name for the disease, it would be prudent to do so separately from establishment of the acronym.

 We now define the COVID-19 acronym in the first sentence of the introduction paragraph.

 Line 179: Please define/explain the use of the term boxcars.

 We have provided a “boxcar” definition, given more motivation for its use, and an example of 1 vs 7 compartments in the methods section: “Similar to the methods in [5], the exposure and infectious periods consisted of “boxcars”, smaller consecutive compartments that each individual must pass through. Boxcars enforce the minimum number of days spent in each compartment and more accurately reflect the waiting time distribution of a negative binomial distribution [41,42]. For example, an infectious period of 9.5 days could be modeled as one compartment with a daily transition rate of 1/9.5 or broken up into seven boxcars with a daily transition rate of 7/9.5.”

 Fig 3: Please define total cases and how this is different from reported cases.

 We agree with the reviewer that the term, “total cases,” was not clear. We now refer to either reported cases or cumulative infections, where cumulative infections includes both the reported and unreported infections. We changed in the text, and within Fig 3 and S7 and their legends.

Reviewer #2

 In this manuscript the authors repurposed a model that was previously used to study the spread of the Zika Virus to estimate the probability of unseen transmission. The model uses a stochastic SEIR branching process to simulate the outbreak until April 2020. The authors find that counties that have confirmed one case have a 50% probability of an epidemic. The authors did a lot of work reconstructing the early stage of the epidemic for each county in the US. Understanding the early phase of an emerging epidemic is crucial and I applaud and thank the authors for their hard work providing situational awareness to both public health departments and the public. With the current version of the paper I have a few issues and concerns about the model assumptions and I believe there is a lack of validation. I have broken down my comments into Major and Minor issues below.

 We thank the reviewer for their thorough comments.

Major comments:

 Model assumptions: A) From my understanding of the paper, it says that all counties have the same Re which is either 1.1, 1.5, or 3 and that is held constant throughout the entire stochastic process. If this is the case, I do not believe this is appropriate. B) In addition, I do not think it is true that all counties had similar transmission rates. I agree with the authors that in early March the Re was most likely closer to 3 due to the lack of interventions. However, it changed when places started social distancing/ working from home/ only opening essential services. There are multiple data sources out there reporting the Rt and changes in mobility (some at the county level) that could be incorporated to adjust for this. I know it is a lot of work but including this heterogeneity could change the results. C) I am also concerned about the estimate for the Latent period, this value (1.25 days) is one of the smaller values I have seen for the early estimates of this parameter. Most of the values I have read are between 2-7 days.

 A) The reviewer is correct that the effective reproduction number is held constant during the whole simulation. We have added text to the methodology and included this as a limitation in our discussion section

 Clarified in our methodology: “Our goal was to estimate the probability that an outbreak in a region would become an epidemic based on the number of observed reported cases in the region and assuming no behavioral changes or public health interventions. As such, we ran 100,000 stochastic outbreak simulations per scenario (Re held constant) beginning with a single undetected case and ending when cumulative infections reached 2,000 or the outbreak died out (whichever came first).”

 As a limitation in our discussion: “Finally, we considered scenarios with lower effective reproduction numbers than estimated (Fig 1 and S3-S7 Figs), which may be more appropriate for the epidemic risks following the enactment of non-pharmaceutical interventions, but we did not account for changes to the effective reproduction number over time. While transmission can vary temporally depending on local policies, testing efforts [33,34], and behavior [13], we made these simplifying assumption, because our overall goal was to estimate county-level epidemic risk in the absence of interventions.”

 B) The reviewer is correct regarding the underlying heterogeneity in spatiotemporal transmission risks. To address these concerns we carried out a completely new analysis that uses previous estimates of the basic reproduction number for each county, rather than assuming the same transmission rates across the country. We call this the retrospective analysis and call the previous analysis our original analysis, since it is based on the best available evidence in March 2020. We now additionally compare the results between the retrospective and original analyses.

 All text and figures from the manuscript have shifted to focus on the retrospective analysis in response to the reviewer's concerns.

 All previous figures were moved to the supplement, and we have replicated all analyses both for the retrospective and original analyses.

 New text from introduction addressing different results: “We present results from a model using the best estimates for COVID-19 epidemiological characteristics as of December 2022 (retrospective) and compare those results with those made in early March 2020 (original).”

 We additionally include a comparison between the original and retrospective analyses in the results: “As additional validation of the modeling framework, we compare the estimates originally made in March 2020, before we had county-specific estimates of reproduction numbers (Table 1, Fig 5). At that time, we assumed all counties had the same effective reproduction numbers, ranging from 1.1 to 3.0; we also originally assumed a latent period of 1.25 rather than 2.9 days. Our original estimates assuming Re=3.0 most closely match the retrospective estimates (Pearson's product-moment correlation, r=0.99; p<0.001). Our original county-level risk maps (S3-S5 Figs) and estimates for the time until counties will reach 1,000 cumulative infections (S6 and S7 Figs) are also consistent with our retrospective analysis. Finally, our original estimates reliably predicted subsequent county case increases (S9 Fig). For example, assuming Re=3.0, a 10% increase in estimated epidemic risk corresponds to an increase in the log odds of a county detecting at least one, two, or five new cases by March 23 of 0.48 (95% CI: 0.43-0.53), 0.49 (95% CI: 0.45-0.53), and 0.55 (95% CI: 0.51-0.59), respectively. Comparing logistic regression models built on the retrospective analysis to the original epidemic risk estimates where Re=3.0, we find that the retrospective risk estimates more accurately predict the probability of a county reporting at least two new reported cases in the week following March 16, 2020 (AIC difference of 93.3 and 100% weight in favor of the retrospective risk estimates).”

 The new Figure 5:

Fig 5. Comparison of original epidemic risk estimates, assuming a uniform Re across counties, and retrospective estimates, assuming empirical county-level estimates of Re on March 16, 2020 across 3,142 US counties. Each point corresponds to a pair of risk estimates (original on x-axis vs retrospective on y-axis) for a single county. Points are shaded according to the assumed effective reproduction number for the original estimate. The solid diagonal line indicates matching estimates.

 New discussion text addressing comparison: “We also compare the results of our analysis with estimates that we originally made on March 16, 2020, before we had data that allowed us to estimate county-specific SARS-CoV-2 reproduction numbers. At that time, we made the simplifying assumption that transmission rates were uniform across counties. Our original estimates are consistent with both our retrospective estimates, though slightly less accurate (Fig 5). Importantly, even the data limited analyses provided clear indication of the extent of undetected epidemic risk and urgency of action across the US.”

 Description in the methodology about the retrospective reproduction number estimates: “For the retrospective analysis, epidemic risk is based on the county-specific cumulative reported cases and county-specific effective reproduction numbers (Re). Epidemiological parameters for the model are drawn from a literature search carried out in December 2022, which updated the best estimate for the COVID-19 latent period from 1.25 to 2.9 days (see comparison in Table 1). We assume that the county Re equals the basic reproduction number estimated in [7] for all counties in the contiguous US. As population density and urban-rural classification are strong predictors for the COVID-19 reproduction number [9,40], we estimated the Re for counties in Alaska and Hawaii as the mean Re of all the contiguous US counties with the same urban-rural designation code as defined by 2013 estimates from the National Center for Health Statistics Urban-Rural Classification Scheme for Counties rounded to the nearest tenth [10]. In total, counties had twenty-nine different Re values ranging from 1.4 to 4.4. We included Re=1.1 as well to simulate a possible social distancing scenario when counties were under shelter-in-place orders [26].”

 C) We agree with the reviewer and now have included a latent period of 2.9 days [13] for our retrospective analysis but have kept the latent period at 1.25 days for the original analysis because it was the best available at the time. We outline the parameter choices in Table 1 and the methodology.

 New Table 1:

Table 1. Model parameters used for simulating COVID-19 outbreaks.

Parameter Description Original Source Retrospective Source

Re Effective reproduction number: Average number of new cases from one infected individual in a susceptible and non-susceptible population 1.5

1.1, 3.0 [8]

[9] Contiguous US: fit to each county

Alaska and Hawaii: mean of urban-rural designation [7]

[10]

TG Generation time (days): Average length of time between consecutive exposures 6 [11] 6 [12]

TE Latent period (days) 1.25 Fit to TG 2.9 [13]

TI Infectious period (days) 9.5 [11] 6.2 Fit to TG

e Number of exposed compartments in boxcar implementation (min days of exposure) 1 floor(TE) 2 floor(TE)

n Number of infectious compartments in boxcar implementation (min days of infectiousness) 7 [11] 6 floor(TI)

 Latency rate: Daily probability of progressing from one exposed compartment to the next 0.80 e/TE 0.69 e/TE

 Recovery rate: Daily probability of progressing from one infectious compartment to the next 0.73 n/TI 0.97 n/TI

 Daily detecting rate: The daily probability of an infectious individual being detected, 0.01 [14] 0.01 [14]

k Total dispersion parameter of negative binomial distribution 0.16 [15] 0.16 [15]

 Daily detection rate: Probability of an on-going infection becoming a reported case 0.1 [4] 0.1 [4]

 R code for number of new infectious individuals drawn daily:

rnbinom(n=1,prob=k/(R_(e )+ k),size=k/T_I ) 

 We updated the methodology to discuss this change: “Epidemiological parameters for the model are drawn from a literature search carried out in December of 2022, which updated the best estimate for the COVID-19 latent period from 1.25 to 2.9 days (see comparison in Table 1).”

 Model calibration: Can the authors also confirm the calibration procedure for each individual county? From what I read, in order to get the variation in the timing of individual counties, stochastic runs are shifted to match the case data. However, I’m not entirely sure exactly how this is done. Do you choose one date (March 16) or a group of dates? I guess I’m not sure what the “fitting window” is and how the model is mapped to actual dates because the model doesn’t use information on importations, the timing of the start of the outbreak… etc.

 Thank you for this comment. We now have included a calibration process example as part of Figure 1 and made a specific section in the methodology to describe it.

 New figure 1 and legend:

Fig 1. Epidemic risk for the effective reproduction numbers (Re) corresponding to reduced risk (1.1) and the minimum (1.4), median (2.8), and maximum (4.4) estimated across all US counties. For a given number of reported cases, epidemic risk increased with estimated Re. By the time a single case was reported, there was a 13%, 45%, 81%, or 89% chance of an ongoing epidemic for an Re of 1.1 (reduced risk), 1.4 (minimum), 2.8 (median), or 4.4 (maximum), respectively. County-specific risk is estimated from these curves. For example, Travis County, TX (red lines) had an Re of 2.0, which corresponds to an epidemic risk of 95% on March 13, 2020 and 99% on March 20, 2020 based on cumulative reported case counts of four and twenty-one on those dates, respectively. If the Re was instead estimated as 1.1 in Travis County, then the estimated risk would decrease to 57% based on the twenty-one cases reported on March 20. The model assumed a 10% case detection rate, generation time of 6.0 days, a latent period of 2.9 days, and infectious period of 6.2 days (Table 1 - retrospective).

 Methodology from the County Epidemic Risk Assignment section: “We matched county cumulative reported case numbers (confirmed and suspected from [19]) with the detected case number to obtain epidemic probabilities for each US county based on their reported cumulative number of cases and through matching the county-specific Re. For the main analysis each county was matched by its estimated Re as described above. For the original analysis presented in the supplement, we explore effective reproduction numbers ranging from 1.1 to 3.0.”

 Validation: A) This paper is missing statistical validation of the model estimates vs. observational data. The model is currently validated by looking at whether case counts increased in the following week. In Figure 4, the model estimates are overlapped with the values using reported cases. However, I think a more quantitative way of validating the model against the surveillance data should be used. Possibly some correlation measure. Another type of validation measure that could be useful is based on the timing of community transmission at the county level (answering “when” did counties have widespread transmission and was it similar to what we observed with reported cases). So looking at the temporal evolution of the early phase. B) I also think that this paper is missing a discussion on how this analysis aligns with the literature. There has been a lot of work on the early phase of the epidemic using phylogenetic analysis, retrospective samples, waste water surveillance, and other epidemic modeling approaches within the US (not much at the county level). While there isn’t a single, comprehensive paper to compare this work to, it would contribute to the validation of the model and deepen the discussion.

 A) We agree that our estimates could use more validation. We completely revamped our analysis and have included multiple layers of retrospective validation within the results, methodology, and discussion. For the complete changes please see our answer to Question 3B above.

 B) We agree that comparisons with the literature are now useful and have added more text to address how our results compare with the current understanding for early COVID-19 dynamics in the US. New paragraph in the discussion: “Our results are consistent with the current understanding of early COVID-19 transmission in the United States. Epidemiological and phylodynamic models identified substantial, undocumented, COVID-19 transmission leading up to stay-at-home orders in late March 2020 [14,25], with non-pharmaceutical interventions reducing transmission and preventing infections and mortality [24,26]. Proactive responses to COVID-19 have been estimated to shorten the duration of costly measures [27,28], whereas delays have likely cost lives [26,29]. Thus, our results suggest that the first reported case should trigger action if the goal is to fully contain an emerging outbreak as quickly as possible. The risk of an ongoing epidemic likely already exceeds 50% and delaying action may substantially reduce the window for corrective action and amassing adequate healthcare and other mitigation resources (Fig 3).”

Minor comments:

 Language of unseen transmission: As of now, the paper uses many different words and phrases to describe an “unseen epidemic” and how to estimate it such as: epidemic risk, unseen transmission, widespread transmission, sustained community transmission, and outbreaks that spread widely. It would useful and helpful to the reader to use a common terminology that is defined explicitly in the main paper. From what I read “epidemic risk” is the most well defined, but I’ll leave that up to the authors to decide what works best.

 We have changed the language throughout the paper to limit the use of phrases other than ‘epidemic risk’ where appropriate. We have kept some instances of “sustained community transmission” as it is used to define the term ‘epidemic’ and ‘undetected epidemic’ to highlight the use of the model for cryptic disease transmission.

---

## [Decision Letter · Decision Letter 1]

23 Mar 2023

Estimating the undetected emergence of COVID-19 in the US

PONE-D-22-17433R1

Dear Dr. Javan,

We’re pleased to inform you that your manuscript has been judged scientifically suitable for publication and will be formally accepted for publication once it meets all outstanding technical requirements.

Kind regards,

Sung-mok Jung

Academic Editor

PLOS ONE

Additional Editor Comments (optional):

I believe all of the comments from reviewers have been well addressed in the revised manuscript. I appreciate authors' effort and congratulations on the publication.

Reviewers' comments:

Reviewer's Responses to Questions

**Comments to the Author**

1. If the authors have adequately addressed your comments raised in a previous round of review and you feel that this manuscript is now acceptable for publication, you may indicate that here to bypass the “Comments to the Author” section, enter your conflict of interest statement in the “Confidential to Editor” section, and submit your "Accept" recommendation.

Reviewer #2: All comments have been addressed

2. Is the manuscript technically sound, and do the data support the conclusions?

Reviewer #2: Yes

3. Has the statistical analysis been performed appropriately and rigorously? 

Reviewer #2: Yes

4. Have the authors made all data underlying the findings in their manuscript fully available?

Reviewer #2: Yes

5. Is the manuscript presented in an intelligible fashion and written in standard English?

Reviewer #2: Yes

6. Review Comments to the Author

Reviewer #2: (No Response)

7. PLOS authors have the option to publish the peer review history of their article (what does this mean?). If published, this will include your full peer review and any attached files.

Reviewer #2: No

---

## [Editor Report · Acceptance letter]

30 Mar 2023

PONE-D-22-17433R1 

Estimating the undetected emergence of COVID-19 in the US 

Dear Dr. Javan:

I'm pleased to inform you that your manuscript has been deemed suitable for publication in PLOS ONE. Congratulations! Your manuscript is now with our production department. 

Kind regards, 

on behalf of

Dr. Sung-mok Jung 

Academic Editor

PLOS ONE